# Dynamics of Magnetized and Magnetically Charged Particles around Regular Nonminimal Magnetic Black Holes

Javlon Rayimbaev [1,2,3,4,5,*], Bakhtiyor Narzilloev [1,2,3], Ahmadjon Abdujabbarov [1,3,4,6,7] and Bobomurat Ahmedov [1,3,7]

- [1] Ulugh Beg Astronomical Institute, Astronomy Street 33, Tashkent 100052, Uzbekistan; nbakhtiyor18@fudan.edu.cn (B.N.); ahmadjon@astrin.uz (A.A.); ahmedov@astrin.uz (B.A.)
- [2] College of Engineering, Akfa University, Kichik Halqa Yuli Street 17, Tashkent 100095, Uzbekistan
- [3] Faculty of Physics, National University of Uzbekistan, Tashkent 100174, Uzbekistan
- [4] Institute of Nuclear Physics, Ulugbek 1, Tashkent 100214, Uzbekistan
- [5] Power Engineering Faculty, Tashkent State Technical University, Tashkent 100095, Uzbekistan
- [6] Shanghai Astronomical Observatory, 80 Nandan Road, Shanghai 200030, China
- [7] Department of Physics, Tashkent Institute of Irrigation and Agricultural Mechanization Engineers, Kori Niyoziy, 39, Tashkent 100000, Uzbekistan
- [*] Correspondence: javlon@astrin.uz

**Abstract:** The present paper is devoted to the study of the event horizon properties of spacetime around a regular nonminimal magnetic black hole (BH), together with dynamics of magnetized and magnetically charged particles in the vicinity of the BH. It is shown that the minimum value of the outer horizon of the extreme charged BH increases with the increase in coupling parameter. It reaches its maximum value of $1.5M$ when $q \to \infty$, while the maximal value of the BH charge decreases and tends toward zero. We also present a detailed analysis of magnetized particles' motion around a regular nonminimal magnetic black hole. The particle's innermost circular stable orbits (ISCOs) radius decreases as the magnetic charge and the parameter $\beta$ increase and the coupling parameter of Yang–Mills field causes a decrease at the values of the magnetic charge near to its maximum. We show that the magnetic charge can mimic the spin of a rotating Kerr black hole up to the value of $a = 0.7893M$, providing the same value for an ISCO of a magnetized particle with the parameter $\beta = 10.2$ when the coupling parameter is $q = 0$. Moreover, Lyapunov exponents, Keplerian orbits and harmonic oscillations of magnetized particles motion are also discussed.

**Keywords:** modified gravity; magnetized particles; magnetic field; center-of-mass energy

**PACS:** 04.50.-h; 04.40.Dg; 97.60.Gb



## 1. Introduction

From the theoretical point of view, the demand for the exploration of alternative theories of gravity is associated with the fundamental problems of classical general relativity, such as the existence of a singularity at the origin of the exact black hole solutions of field equations and the fact that it is not compatible with quantum field theory. On the other hand, the current rich experimental and observational data, in principle, could justify the general relativity in the strong gravity regime. However, the accuracy of the experiments and observations does not provide a final answer on the validity of general relativity and leaves an open window to consider other alternative and modified theories of gravity.

In the literature, there is a large number of alternative and modified theories of gravity proposed to describe different astrophysical processes and the nature of exotic mysterious objects. In turn, this creates new problems associated with dealing with the set of a large number of parameters and obtaining constraints on different theories of gravity using observational data. One of the attempts to resolve this issue has been the parametrization

of the spacetime metric surrounding the astrophysical compact object and comparing them with the parameters of alternate gravity theories [1–3].

Since the electromagnetic field (and other scalar fields associated with them) is an essential part of spacetime surrounding astrophysical compact objects, one may consider the theories that couple the gravitational field to other ones to construct solutions within alternative theories of gravity. One may distinguish the following five types of nonminimal field theories coupled to gravity:

- Models dealing with scalar fields coupled to gravity. The so-called Scherer–Jordan–Thiry–Brans–Dicke theory has been widely studied in Refs. [4–9], including the study of scalar fields conformally coupled to gravity.
- The Einstein–Maxwell model based on the nonminimal coupling of the electromagnetic field with gravity. The review of this type of theory can be found in Refs. [10–13].
- Models with SU(n) symmetry, usually known as Einstein–Yang–Mills theories, have been reviewed in Ref. [14].
- So-called Einstein–Yang–Mills–Higgs models have been explored in Refs. [15,16].
- The Einstein–Maxwell-axion models are related to the axion pseudoscalar field coupled with either the electromagnetic or gravitational field [17].

The solutions describing the spacetime surrounding the compact relativistic objects within nonminimally coupled gravity theories have been obtained in Refs. [18–26]. The different black hole solutions within the Einstein–Yang–Mills theory have been explored in Refs. [27–32]. Particularly, black hole solutions, including Lorentz group symmetry and loop quantum gravity in the Einstein–Yang–Mills theory, have been found in Refs. [33,34]. Here, we plan to explore the spacetime structure and test particle dynamics around black holes described by the solution of the Einstein–Yang–Mills theory coupled with the SU(2) gauge field [25,35,36].

The electromagnetic field structure and charged/magnetized particle motion around a black hole can be considered as a useful instrument to explore the gravity theory in a strong-field regime. Particularly, the solution of the electromagnetic field equation around a black hole immersed in an external asymptotically uniform magnetic field has been obtained in [37]. The modification of the astrophysical processes due to the electromagnetic field has been explored in Refs. [38–43]. A large number of studies have been devoted to the study and analysis of the spacetime structure and particle dynamics around black holes in the presence of external electromagnetic fields (see, e.g., Refs. [44–92]).

Magnetized particles' motion in spacetime around a compact object in the presence of an external electromagnetic field may be also applied to model various astrophysical scenarios [93–110].

Here, we aimed to explore the spacetime structure by studying neutral, magnetically charged, and magnetized particle dynamics around a black hole within the Einstein–Yang–Mills theory. The paper is organized as follows: in Section 2, we review the spacetime metric and its properties. We study the magnetized particle motion in Section 3 and magnetically charged particle dynamics in Section 4. We conclude the results of the paper in Section 5. Throughout the paper, we use the system of units where $G = 1 = c$. Greek (Latin) indices run from $0(1)$ to 3.

## 2. The Spacetime Properties

The action of the non-minimally coupled Einstein–Yang–Mills theory in four dimensional spacetimes has the following form [25,35]:

$$\mathcal{S} = \int d^2x \sqrt{-g} \left[ \frac{R}{8\pi} + \frac{1}{2} \left( F_{\mu\nu} F^{\mu\nu} + \mathcal{R}^{\alpha\beta\mu\nu} F_{\alpha\beta} F_{\mu\nu} \right) \right], \tag{1}$$

where $g$ is the determinant of the metric tensor and $R$ is the Ricci scalar. The Yang–Mills (YM) tensor, $F_{\mu\nu}$, is connected to the YM potential, $A_\mu$, by the following relation:

$$F_{\mu\nu} = \Delta_\mu A_\nu - \Delta_\nu A_\mu + \kappa A_\mu A_\nu, \tag{2}$$

where $\Delta_\mu$ is the covariant derivative and $\kappa$ is the structure constant parameter of the Yang–Mills field. The Riemann tensor for the Yang–Mills field given in [35] takes the following form under the Wu–Yang ansatz (see [36]) :

$$
\begin{aligned}
\mathcal{R}^{\alpha\beta\mu\nu} =\ & -\frac{q}{2}\Big\{12R^{\alpha\beta\mu\nu} + g^{\alpha\mu}g^{\beta\nu} - g^{\alpha\nu}g^{\beta\mu} \\
& + R^{\alpha\mu}g^{\beta\nu} - R^{\alpha\nu}g^{\beta\mu} + R^{\beta\nu}g^{\alpha\mu} - R^{\beta\mu}g^{\alpha\nu}\Big\},
\end{aligned}
\tag{3}
$$

where $R^{\alpha\beta}$ is the Ricci tensor and $q$ is the minimally coupled parameter between the Yang–Mills field and the gravitational field.

The spacetime around magnetically charged regular nonminimal magnetic black holes can be expressed as [25,35,36]

$$
ds^2 = -f(r)dt^2 + \frac{1}{f(r)}dr^2 + r^2(d\theta^2 + \sin^2\theta\, d\phi^2)\,,
\tag{4}
$$

with the lapse (radial) function

$$
f(r) = 1 + \left(1 + \frac{2Q_m^2 q}{r^4}\right)^{-1}\left(-\frac{2M}{r} + \frac{Q_m^2}{r^2}\right),
\tag{5}
$$

where $M$ and $Q_m$ are the total mass and magnetic charge of the black hole, respectively.

The electromagnetic field four-potential of the non-rotating regular nonminimal magnetic black hole reads

$$
A_\alpha = (0,0,0,Q_m\cos\theta)\,.
\tag{6}
$$

In fact, when $q = 0$, the spacetime metric (5) describes the spacetime around magnetically charged Reissner–Nordström black holes.

Now, we explore the properties of the event horizon of the regular nonminimal magnetic BH spacetime governed by the lapse function (5) and see how it depends on the nonminimal parameter, $q$, and the regular BH charge, $Q_m$.

Figure 1 shows the radial dependence of the lapse function of the extreme charged regular nonminimal magnetic BH for the different values of the nonminimal coupling parameter.

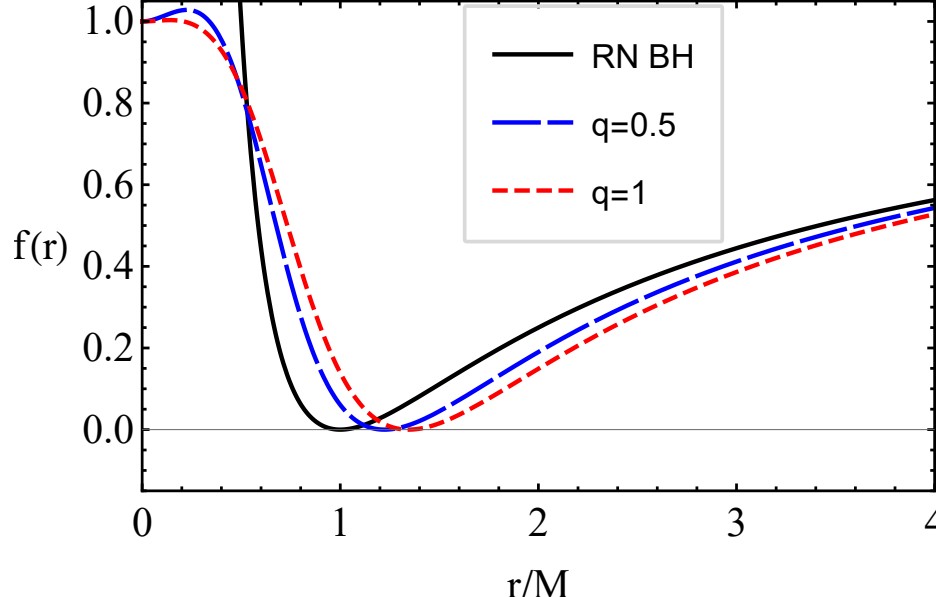

**Figure 1.** The radial dependence of the lapse function for the different values of the nonminimal parameter and the corresponding values of the magnetic charge $Q_m \to Q_{extr}$.

Generally, the radius of the event horizon of a BH is described by the standard way, setting $g_{rr} \to \infty$, $g^{rr} = 0$ or, equivalently, through the solution of the equation

$$f(r) = 0\,, \tag{7}$$

from which we have

$$
\left(\frac{r_h}{M}\right)_{\pm} = 1 + Y_1 \pm \left\{ 2 + \frac{2}{Y} - \frac{Y_1}{3\sqrt[3]{2}M^2} \right.
$$
$$
\left. - \frac{Q_m^2}{M^2}\left(\frac{4}{3} - \frac{2}{Y} + \frac{\sqrt[3]{2}(24q + Q_m^2)}{3Y_1}\right) \right\}^{\frac{1}{2}}, \tag{8}
$$

where

$$
\frac{1}{2}Y^3 = 108M^2qQ_m^2 - 72qQ_m^4 + Q_m^6 \tag{9}
$$
$$
+ \sqrt{\left(108M^2qQ_m^2 - 72qQ_m^4 + Q_m^6\right)^2 - \left(24qQ_m^2 + Q_m^4\right)^3}\,,
$$
$$
Y_1^2 = 1 - \frac{2Q_m^2}{3M^2} + \frac{Y}{3\sqrt[3]{2}} + \frac{\sqrt[3]{2}Q_m^2}{3Y}\left(24q + Q_m^2\right),
$$

and $\pm$ stands for the outer and inner horizons.

One can see the effects of the nonminimal coupling parameter, $q$, and the BH charge, $Q_m$, on the event horizon in Equation (8).

The dependence of the event horizon of the regular nonminimal magnetic BH from the BH charge is shown in Figure 2 for the different values of the nonminimal coupling parameter. One can see from the figure that an increase in the magnetic charge of a BH increases the inner horizon radius and decreases the outer one. It is also clearly seen from the graph that for specific values of the nonminimal coupling parameter, we gain corresponding values of the magnetic charge parameter, causing the inner and outer horizons to meet each other.

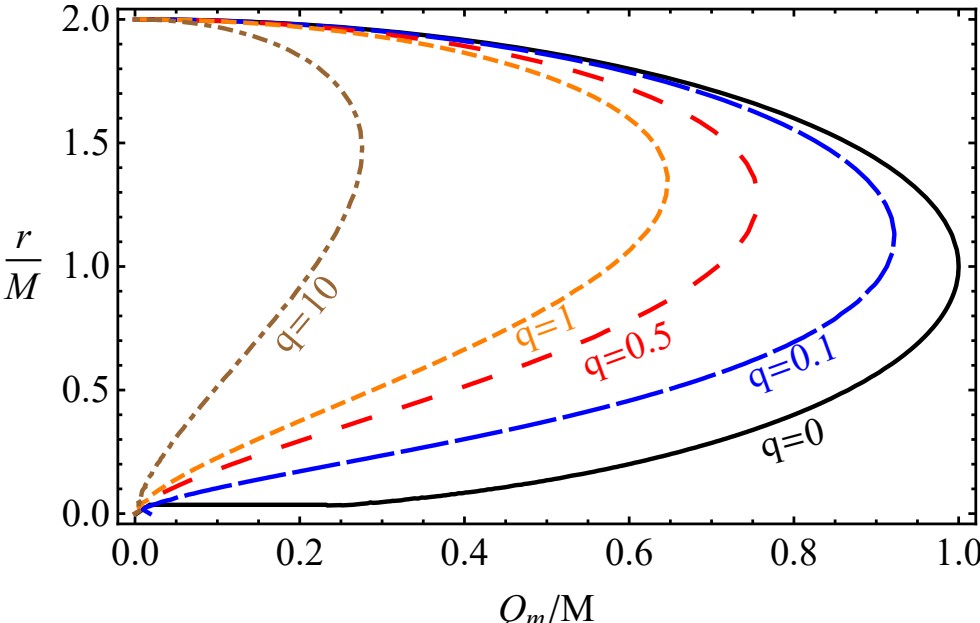

**Figure 2.** The dependence of event horizon radius from the magnetic charge of the regular magnetic BH – $Q_m$, for the different values of the nonminimal coupling parameter, $q$. In the figure, $q/M^2$ is presented as $q$.

The extreme value of the BH charge and minimal value of the event horizon can be easily found by setting the following system of equations and solving them with respect to $r$ and $Q_m$:

$$f(r) = 0 = f'(r) , \tag{10}$$

from which we have

$$\frac{3(r_h)_{min}}{M} = 1 + X + X^{-1}(1 - 12q) \tag{11}$$

$$\begin{aligned}\frac{3Q^2_{extr}}{M^2} &= X^{-2}\left\{1 - 96q^2 + q\left(16X^2 - 62X + 121\right)\right. \\ &+ \left. X^2 + X - 2XX_1 + 5X_1\right\} ,\end{aligned} \tag{12}$$

where

$$X^3 = 1 + 3(21q + X_1), \quad X_1^2 = 3q(q+2)(64q+3) . \tag{13}$$

One may see the effect of the coupling parameter, $q$, on the extreme value of the magnetic BH and the minimal value of its outer horizon using Equations (11) and (12).

One can now show the dependence of the extreme values of the BH charge and minimal value of the outer event horizon from the coupling parameter, as shown in Figure 3. One can see from Figure 3 that when the nonminimal coupling parameter is $q = 0$ (pure RN BH case), the extreme value of the charge and the minimal value of the event horizon are equal to each other and become $Q_{extr} = (r_h)_{min} = M$. One may see that the minimal value of the event horizon increases with the increase in the coupling parameter, while the extreme charge decreases. The limits of the minimal outer horizon and the magnetic charge of the magnetic BH for the infinite value of the coupling parameter can be written as

$$\lim_{q\to\infty} \frac{r_{min}}{M} = \frac{3}{2}, \quad \lim_{q\to\infty} \frac{Q_{extr}}{M} = 0. \tag{14}$$

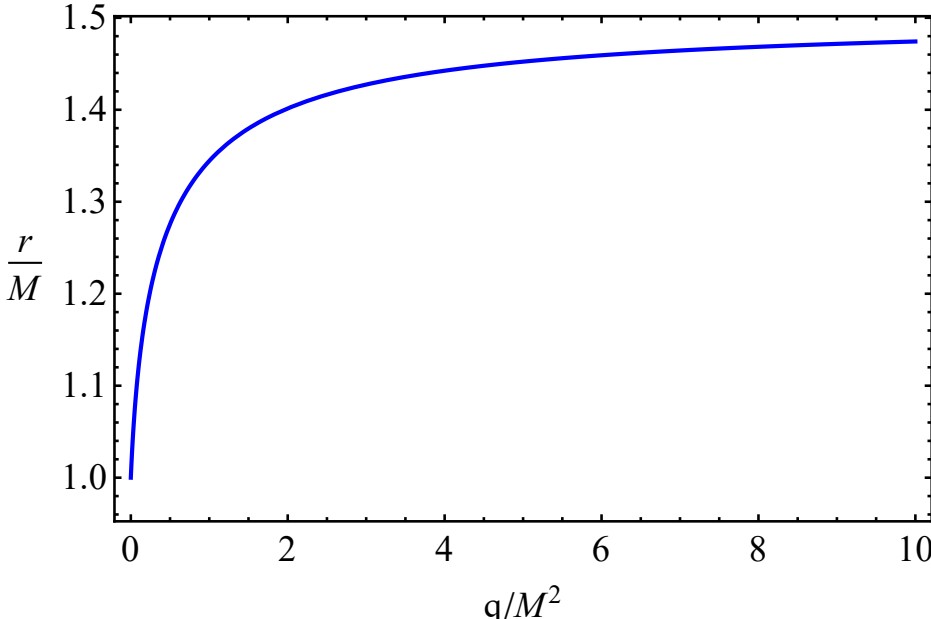

**Figure 3.** *Cont.*

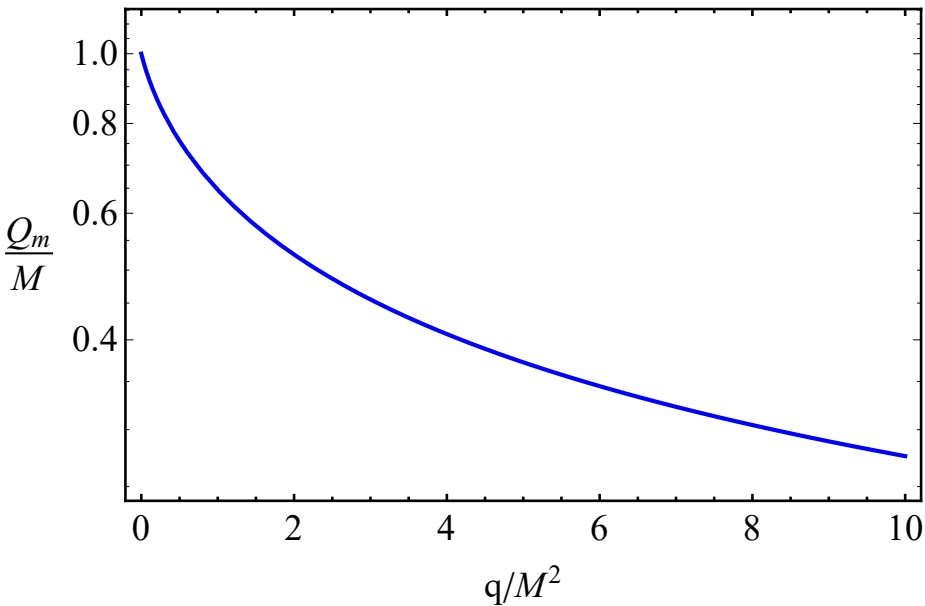

**Figure 3.** The dependence of the minimum value of outer (event) horizon radius (on the **top panel**) and maximum value of the magnetic charge, $Q_m$ (in the **bottom panel**), from the nonminimal coupling parameter, $q$, which allows the existence of BH.

However, in real astrophysical conditions, such a parameter can take limited values and the given expressions above should be considered as the mathematical demonstration of the limits of the given quantities.

## 3. Magnetized Particles Motion

In this section, we focus on the dynamics of magnetized particles with magnetic dipole moment around regular nonminimal magnetic BH.

### 3.1. Magnetic Interaction

One can easily calculate the orthonormal radial component of the magnetic field generated by the magnetic charge of the RN-BH using the electromagnetic four-potentials given in Equation (6) in the following form:

$$B^{\hat{r}} = \frac{Q_m}{r^2} \, . \tag{15}$$

The radial component of the magnetic field around a magnetically charged black hole formally coincides with the standard Newtonian expression.

The dynamics of a magnetized particle around a regular nonminimal magnetic black hole can be studied using the following Hamilton–Jacobi Equation [93]

$$g^{\mu\nu} \frac{\partial \mathcal{S}}{\partial x^\mu} \frac{\partial \mathcal{S}}{\partial x^\nu} = -\left( m - \frac{1}{2} \mathcal{D}^{\mu\nu} F_{\mu\nu} \right)^2 , \tag{16}$$

where the term $\mathcal{D}^{\mu\nu} F_{\mu\nu}$ stands for the interaction between the magnetic dipole moment of the magnetized particle, $\mu^\nu$, and the magnetic field generated by the magnetic charge of the regular nonminimal magnetic black hole. We assume that the magnetized dipole moment of the magnetized particle has to be satisfied by the following condition with the corresponding polarization tensor, $\mathcal{D}^{\alpha\beta}$:

$$\mathcal{D}^{\alpha\beta} = \eta^{\alpha\beta\sigma\nu} u_\sigma \mu_\nu \, , \qquad \mathcal{D}^{\alpha\beta} u_\beta = 0 \, . \tag{17}$$

The interaction term $\mathcal{D}^{\mu\nu}F_{\mu\nu}$ can be determined by the relation between the electromagnetic field tensor, $F_{\alpha\beta}$, which is expressed through the components of electric, $E_\alpha$, and magnetic, $B^\alpha$, fields in the following form:

$$F_{\alpha\beta} = w_\alpha E_\beta - w_\beta E_\alpha - \eta_{\alpha\beta\sigma\gamma} w^\sigma B^\gamma \ . \tag{18}$$

Taking into account the condition given in Equation (17) and non-zero components of the electromagnetic field tensor, one can write the interaction term in the following form:

$$\mathcal{D}^{\alpha\beta} F_{\alpha\beta} = 2\mu_\alpha B^\alpha = 2\preceq^{\hat{\alpha}} B_{\hat{\alpha}} \ . \tag{19}$$

It is known that the magnetic interactions are in the equilibrium state with minimal interaction energy when the directions of magnetic field lines and the magnetic dipole moment of the magnetized particle are parallel to each other. For this case, the direction of the magnetic dipole moment of the magnetized particle lies at the equatorial plane, being parallel to the magnetic field generated by the magnetic charge of the regular nonminimal magnetic RB BH with the orthonormal components, $\mu^{\hat{i}} = (\mu^{\hat{r}}, 0, 0)$. One may now rewrite the magnetic interaction using Equations (17) and (19) in the following form:

$$\mathcal{D}^{\alpha\beta} F_{\alpha\beta} = \frac{2\mu Q_m}{r^2} \ . \tag{20}$$

where $\mu = \left( \left| \mu_{\hat{i}} \mu^{\hat{i}} \right| \right)^{1/2}$ is the norm of the magnetic dipole moment of the magnetized particle.

Now, after obtaining the exact expression for the interaction term in Equation (16), one may analyze the dynamics of the magnetized particles around the regular nonminimal magnetic BH. Since the magnetic field generated by the magnetic regular BH does not violate the axially symmetric spacetime, we still have two conserved quantities of motion of the magnetized particles: energy, $p_t = -E$, and angular momentum, $p_\phi = L$, which allows using the following form of the action:

$$S = -Et + L\phi + S_\theta + S_r \ . \tag{21}$$

Using (19), (16) and the action (21), the radial part of motion of a magnetized particle around a magnetically charged regular nonminimal black hole at the equatorial plane (where $\theta = \pi/2$ and $p_\theta = 0$) can be written in the following form:

$$\dot{r}^2 = \mathcal{E}^2 - V_{\text{eff}}(r; \mathcal{L}, Q_m, q, \beta) \ , \tag{22}$$

where the effective potential has the form:

$$V_{\text{eff}}(r; \mathcal{L}, Q_m, q, \beta) = \left[ \left( 1 - \beta \frac{MQ_m}{r^2} \right)^2 + \frac{\mathcal{L}^2}{r^2} \right]$$
$$\times \left[ 1 + \left( 1 + \frac{2Q_m^2 q}{r^4} \right)^{-1} \left( -\frac{2M}{r} + \frac{Q_m^2}{r^2} \right) \right] , \tag{23}$$

with specific energy, $\mathcal{E} = E/m$, and angular momentum, $\mathcal{L} = L/m$, together with the parameter $\beta = \mu/(mM)$ being responsible for the interaction between the magnetic dipole moment of the magnetized particle and the magnetic charge of the central gravitating object. In the case of a typical neutron star with the dipole magnetic moment, $\mu = (1/2)BR^3$,

treated as a magnetized particle orbiting an SMBH, the parameter $\beta$ takes the following value:

$$
\begin{aligned}
\beta \;\simeq\;\; & 0.128\left(\frac{B}{10^{12}\mathrm{G}}\right)\left(\frac{R}{10^6\mathrm{cm}}\right) \\
\times\;\; & \left(\frac{m}{1.4 M_\odot}\right)^{-1}\left(\frac{M}{10^6 M_\odot}\right)^{-1},
\end{aligned}
\tag{24}
$$

where $B$, $m$, and $R$ are the surface magnetic field, mass, and radius of the neutron star, respectively, and $M$ is the mass of the central SMBH. Now, one may evaluate the value of the interaction parameter $\beta$ for the magnetar SGR (PSR) J1745–2900 with magnetic dipole moment $\mu \simeq 1.6 \times 10^{32}\mathrm{G}\cdot\mathrm{cm}^3$ and mass $m \simeq 1.41 M_\odot$ orbiting the SMBH Sgr A* with the mass $M \simeq 3.8 \times 10^6 M_\odot$ at the center of the Milky Way [111] as:

$$
\beta = \frac{\mu_{\mathrm{PSR\,J1745-2900}}}{m_{\mathrm{PSR\,J1745-2900}} M_{\mathrm{SgrA*}}} \approx 10.2 \,.
\tag{25}
$$

In our further analysis of magnetized particle dynamics, we use an astrophysically relevant value for the parameter $\beta = 10.2$, treating the magnetar (SGR) PSR J1745-2900 as a magnetized test particle in SMBH Sgr A* environment.

### 3.2. Stable Circular Orbits

Now, we study the effects of the magnetic charge of the regular BH and the minimal coupling parameters on the ISCO radius of magnetized particles with the parameter $\beta > 0$ around the regular nonminimal magnetic BH.

In order to describe the circular stable orbits of the magnetized particles around the central BH, we use the following standard conditions:

$$
V_{\mathrm{eff}} = \mathcal{E}^2, \qquad V'_{\mathrm{eff}} = 0\,, \qquad V''_{\mathrm{eff}} \geq 0 \,.
\tag{26}
$$

From the first equation in Equation (26), one can find the specific angular momentum and energy of the magnetized particle for circular orbits in the following form:

$$
\begin{aligned}
\mathcal{L}^2 \;=\;\; & \frac{r^2 - \beta M Q_m}{2r^5 Q_m^2(Mq + 2qr + r^3) + 4q^2 r^2 Q_m^4 + r^9(r - 3M)} \\
\times\;\; & \left\{ Q_m\left(Q_m\left[Q_m\left(2q Q_m\left[\beta M Q_m\left(4q + r^2\right) + r^4\right]\right.\right.\right.\right. \\
+\;\; & \left.\beta M r^3\left[8qr + 3r^3 - 2Mq\right]\right) - r^5\left(6Mq + r^3\right)\right] \\
+\;\; & \left.\left.\beta M r^7(2r - 5M)\right) + M r^9\right\},
\end{aligned}
\tag{27}
$$

and

$$
\mathcal{E}^2 = \frac{\left[Q_m^2(2q + r^2) + r^3(r - 2M)\right]^2(r^4 - \beta^2 M^2 Q_m^2)}{2r^7 Q_m^2(Mq + 2qr + r^3) + 4q^2 r^4 Q_m^4 + r^{11}(r - 3M)}.
\tag{28}
$$

The dependence of specific angular momentum and energy of magnetized particles on the parameter $\beta = 10.2$ at circular orbits around magnetically charged regular nonminimal BHs is plotted in Figure 4 for the different values of the nonminimal coupling parameter and fixed values of the magnetic charge of the central object. One can see from the figure that the increase in both the magnetic charge of the BH and the nonminimal coupling parameter causes a decrease in the minimum value of the angular momentum, while the energy's minimum increases. Moreover, the energy and angular momentum exhibit minimum decreases in the presence of non-zero BH charge and the coupling parameter, as is seen in Figure 5.

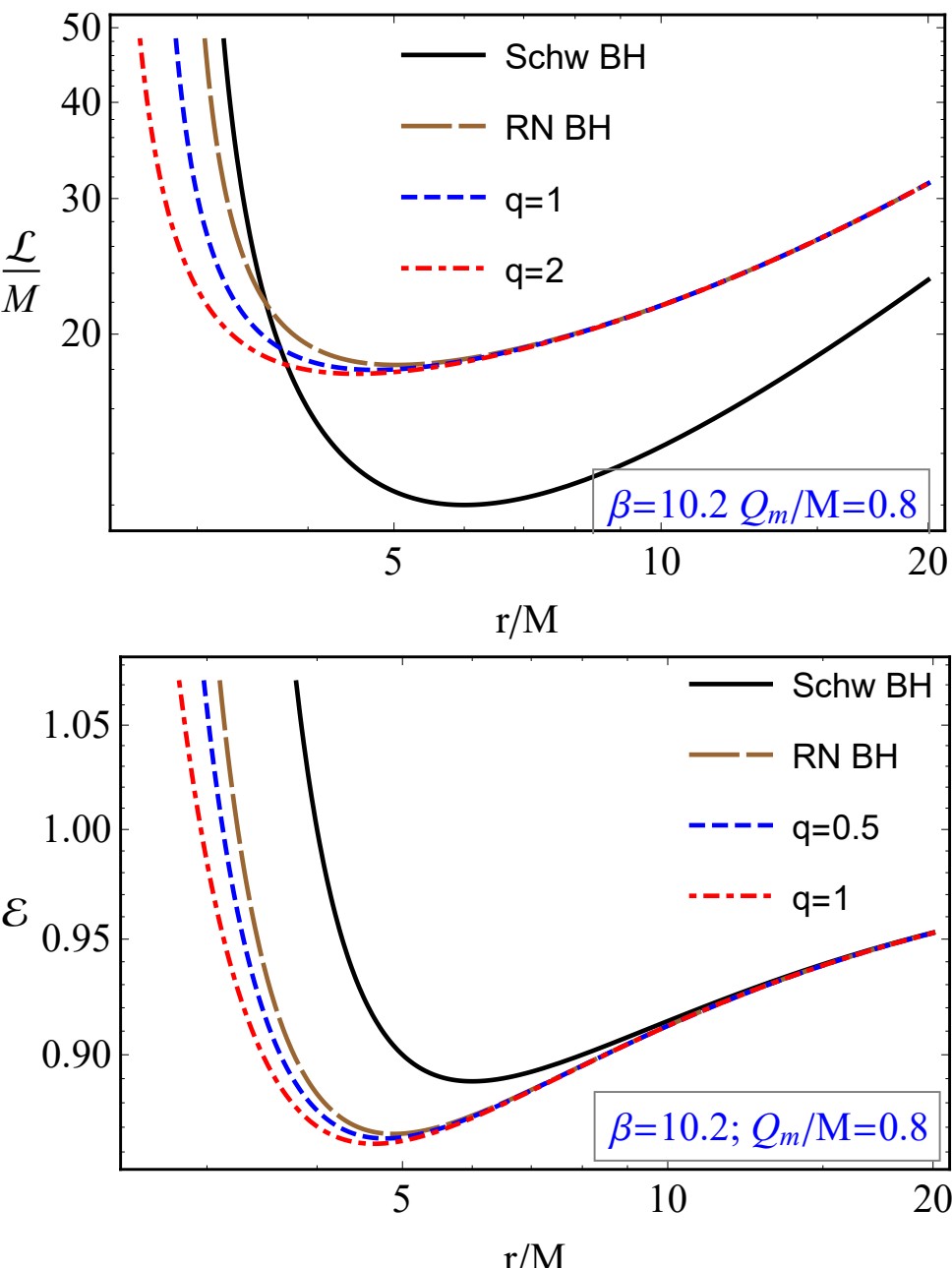

**Figure 4.** The radial dependence of the specific angular momentum (**top panel**) and energy (**bottom panel**) of the magnetized particles for circular orbits around the regular nonminimal magnetic BH for the different values of the nonminimal coupling parameter and fixed values of magnetic charge of the BH and the parameter $\beta = 10.2$.

The possible values of specific energy and angular momentum of magnetized particles, with the parameter $\beta = 10.2$, around magnetically charged BHs for stable and unstable circled orbits are presented in Figure 6. One can see from the figures that the energy of the particles decreases due to the existence of magnetic interaction between the magnetic dipoles and the magnetic field created by the charged BH. The comparison of the extremely charged RN and the regular magnetic BHs show that the energy at the ISCO of particles around RN BH is smaller than the magnetic BH one for a neutral particle, while for magnetized particles, it is bigger for RN BH than the regular magnetic BH due to the extreme charge of the former.

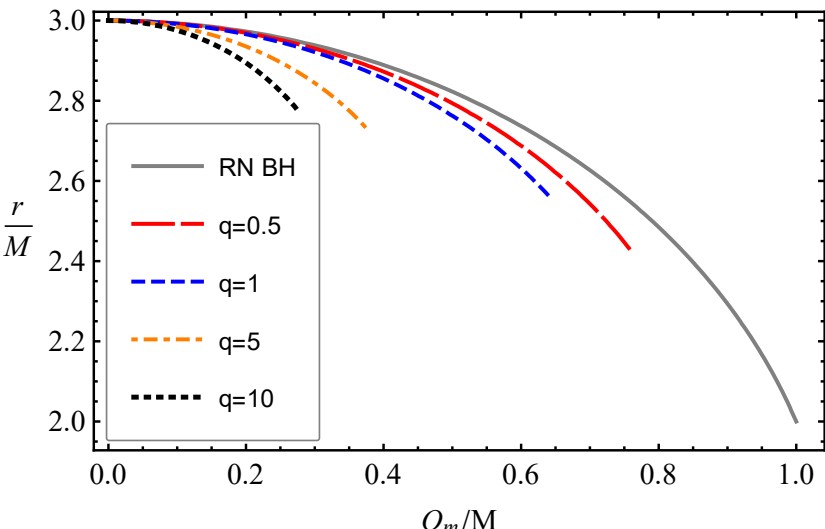

**Figure 5.** The dependence of the minimum distance for the magnetized particles corresponding to circular orbits around the regular nonminimal magnetic BH from the BH charge for the different values of the nonminimal coupling parameter.

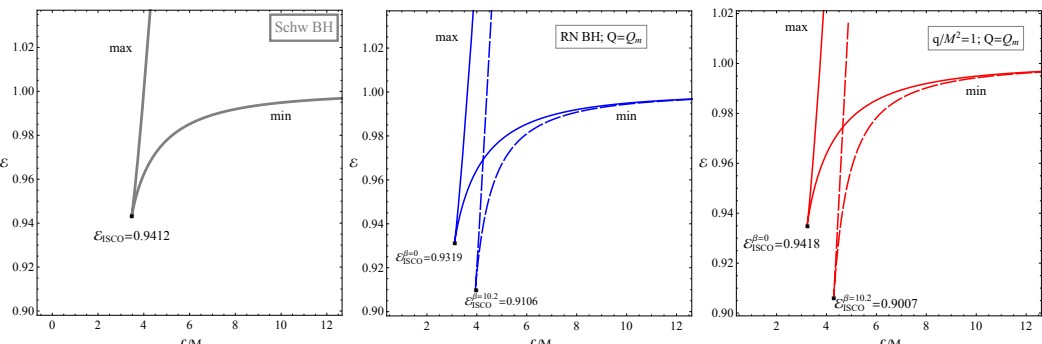

**Figure 6.** Relations between the specific energy and angular momentum of the magnetized particles, with the parameter $\beta = 10.2$, corresponding to stable (corresponding to the minimum of effective potential) and unstable (corresponding to the maximum of effective potential) orbits around an extremely charged regular nonminimal magnetic BH. The comparisons between Schwarzschild and extremely charged RN BHs are provided. The values of energy and angular momentum are located at the ISCO radius, where lines of max and min are connected.

Now, one may study the ISCO radius of the magnetized particles around regular nonminimal magnetic BH using the standard conditions given in Equation (26), producing the following equation:

$$
\begin{aligned}
& 64\beta^2 M^2 q^3 Q_m^8 + 4r^3 Q_m^6 \Big\{ \beta^2 M^2 \Big[ Mq(22q + 3r^2) \\
& + 24q^2 r + 28qr^3 + 6r^5 \Big] + 8q^2 r^3 \Big\} + 2Mr^{10} Q_m^2 \\
& \times \Big[ \beta^2 M (30M^2 - 21Mr + 4r^2) - 12Mq \\
& + 9(4qr + r^3) \Big] - 2r^6 Q_m^4 \Big\{ \beta^2 M^2 \Big[ 4M^2 q \\
& + M(84qr + 37r^3) - 12(2qr^2 + r^4) \Big] + 2Mqr \\
& \times (30q - r^2) + 4(6qr^4 + r^6) \Big\} + 2Mr^{14}(r - 6M) \geq 0 ,
\end{aligned}
\tag{29}
$$

providing the solution for the radial coordinate, which gives the ISCO radius.

One may easily see that Equation (29) is quite complicated and impossible to solve analytically. Hence, we provide analysis of the effects of the magnetic charge of the BH and the nonminimal coupling parameter on the ISCO radius of the magnetized particles numerically with plots.

Figure 7 illustrates the effects of the magnetic charge and the nonminimal coupling parameter on the ISCO radius of magnetized particles around the regular nonminimal magnetic black hole. In the top panel of the figure, we fixed the nonminimal coupling parameter as $q/M^2 = 1$, and one can see that an increase in the magnetic charge and the parameter $\beta$ decreases the ISCO radius. There is an upper limit for the value of the parameter $\beta$ that allows the ISCO to exist, and such an upper value decreases as the magnetic charge grows.

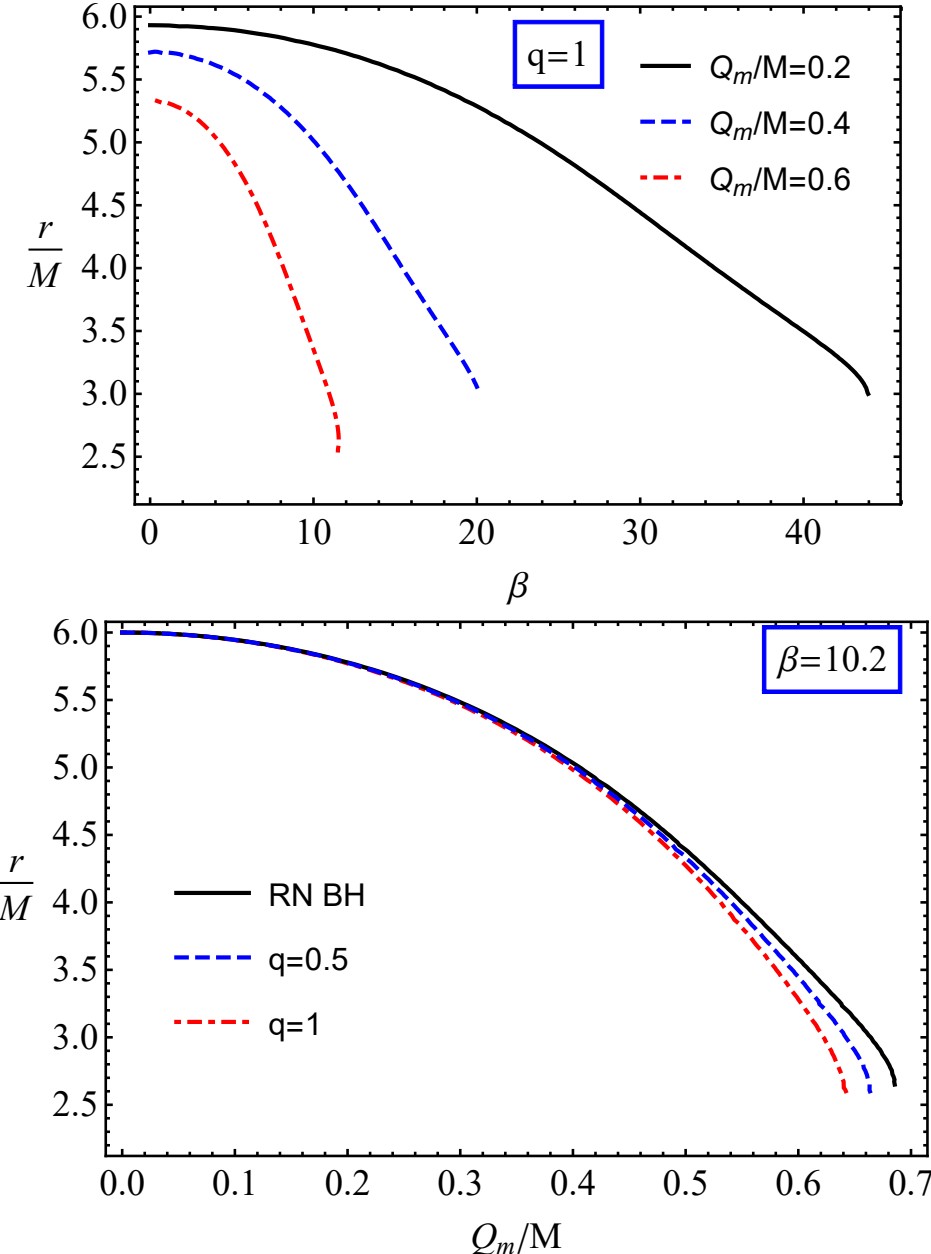

**Figure 7.** The dependence of the ISCO radius of magnetized particles around regular nonminimal magnetic black holes from the magnetic charge of the BH for different values of the magnetic charge (on the **top panel**) and the nonminimal coupling parameter (on the **bottom one**), for the fixed value of the parameter $\beta = 10.2$.

### 3.3. Kerr BH versus Regular Nonminimal Magnetic BH

It is extremely difficult to distinguish the effects of the spin of a central black hole through the measurements of ISCO radii of test particles from observational data of BHs. In most cases, astrophysical black holes are treated as rotating black holes and consequently, a question arises as to whether effects of the BH charge and spin parameters on ISCO are similar. In the other words, how can we be sure that a central BH is charged or spinning? Here, we aimed to provide simple calculations on ISCO radii and to show a new way to distinguish the effects of spin from a magnetic charge of different BHs.

Now, we concentrate on the possibility of distinguishing the effects of magnetic charge of the regular nonminimal magnetic black holes and rotating Kerr black holes through the investigation of the dynamics of magnetized particles, assuming that a magnetized test particle has the same magnetic parameter $\beta = 10.2$ as the magnetar (SGR) PSR J1745-2900 orbiting around SMBH SgrA*.

In Figure 8, we show the degeneracy values of the magnetic charge of the regular nonminimal magnetic BH and spin of the Kerr BH for the different values of the nonminimal coupling parameter. The derivation of the degeneracy between these two spacetime parameters is based on the idea that both the Kerr BH spacetime and the spacetime around a regular nonminimal magnetic BH can have the same ISCO location that coincides with the inner edge of the accretion disk around a BH. One can see that the charge of the pure magnetically charged RN BH ($q = 0$) can mimic the spin parameter up to $a/M = 0.7893$ with its value, $Q_m/M \in (0, 0.645)$, while at $q/M^2 = 1$, the BH charge mimics up to $a/M = 0.82$.

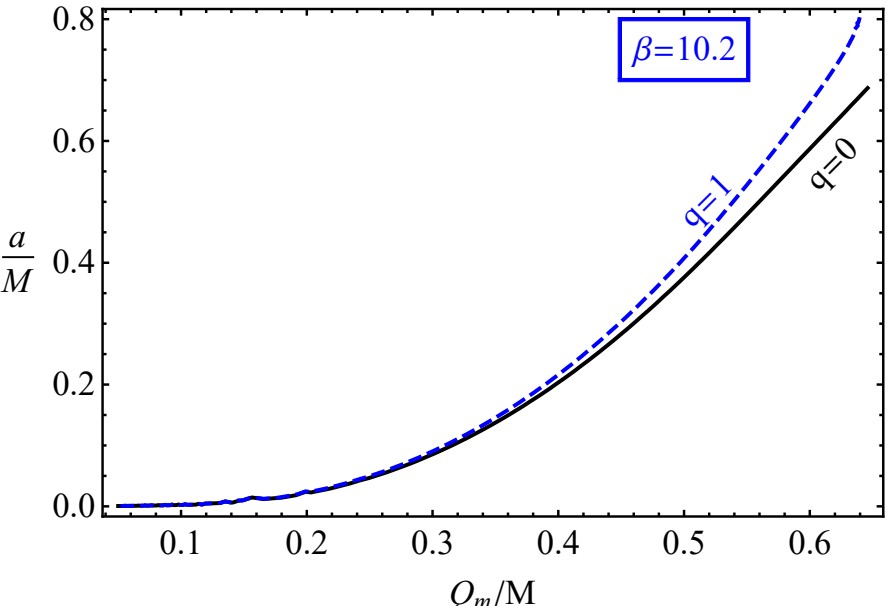

**Figure 8.** Relation between the magnetic charge of a regular nonminimal magnetic black hole and spin of rotating Kerr black hole providing the same value of ISCO radius for the magnetized particle with the parameter $\beta = 10.2$. Here, the unit of the coupling parameter, $q$, is given in $M^2$.

### 3.4. Instability of Circular Orbit

Here, we study the instability of circular orbits for magnetized particles around the magnetic BH by using the Lyapunov exponent describing the measurement of the average rate at which nearby trajectories converge or diverge in the phase space. In other words, the Lyapunov characteristic exponent of a dynamical system is a quantity that characterizes the rate of separation of infinitely close trajectories. A negative Lyapunov exponent designates the convergence between nearby trajectories. A positive Lyapunov exponent determines the divergence between nearby geodesics in which the path of such a system is the most active to change the starting circumstances. A vanishing Lyapunov

exponent designates the existence of marginal stability. Geodesic stability analysis in terms of Lyapunov exponents begins with the equations of motion, schematically written as [112,113]:

$$\lambda = \sqrt{\frac{-\partial_{rr}V_{\text{eff}}(r;\mathcal{L},Q_m,q,\beta)}{2\dot{t}^2}}$$
$$= \frac{2f(r)-rf'(r)}{4r^2(r^4-\beta^2M^2Q_m^2)}\left\{r^2f''(r)\left[\left(r^2-\beta MQ_m\right)^2\right.\right.$$
$$\left.+r^2\mathcal{L}\right]-4rf'(r)\left[r^2\mathcal{L}-2\beta MQ_m\left(r^2-\beta MQ_m\right)\right]$$
$$\left.-4\beta MQ_mf(r)\left(3r^2-5\beta MQ_m\right)+6r^2\mathcal{L}f(r)\right\}^{\frac{1}{2}} \tag{30}$$

The radial dependence of the Lyapunov exponent for neutral (top panel) and magnetized particles (bottom panel) around magnetically charged BHs is presented in Figure 9. It is seen from the figure that the distance where unstable orbits become stable for neutral particles shifts towards the central BH due to the presence of the magnetic charge of the BH. Moreover, one can see that the distance is closer to the central object in the case of the RN BH than the regular magnetic BH one.

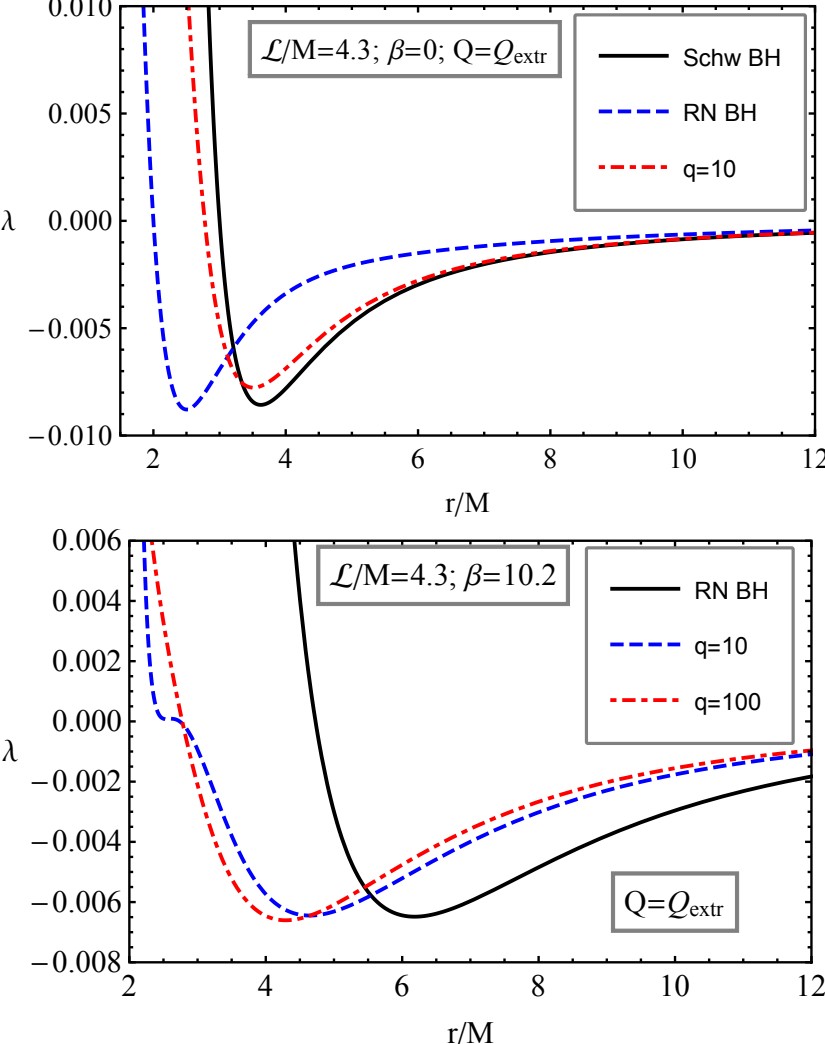

**Figure 9.** The radial dependence of Lyupanov exponent for neutral and magnetized particles around the regular nonminimal magnetic black hole for the various values of the coupling parameter, *q*.

Now, we plan to analyze possible values of the Lyapunov exponent for magnetized particles at the fixed orbits. Here, we provide the analysis graphically in Figure 10 for the different values of the parameter $\beta$ at the distance $r = 3M$ from the central BH.

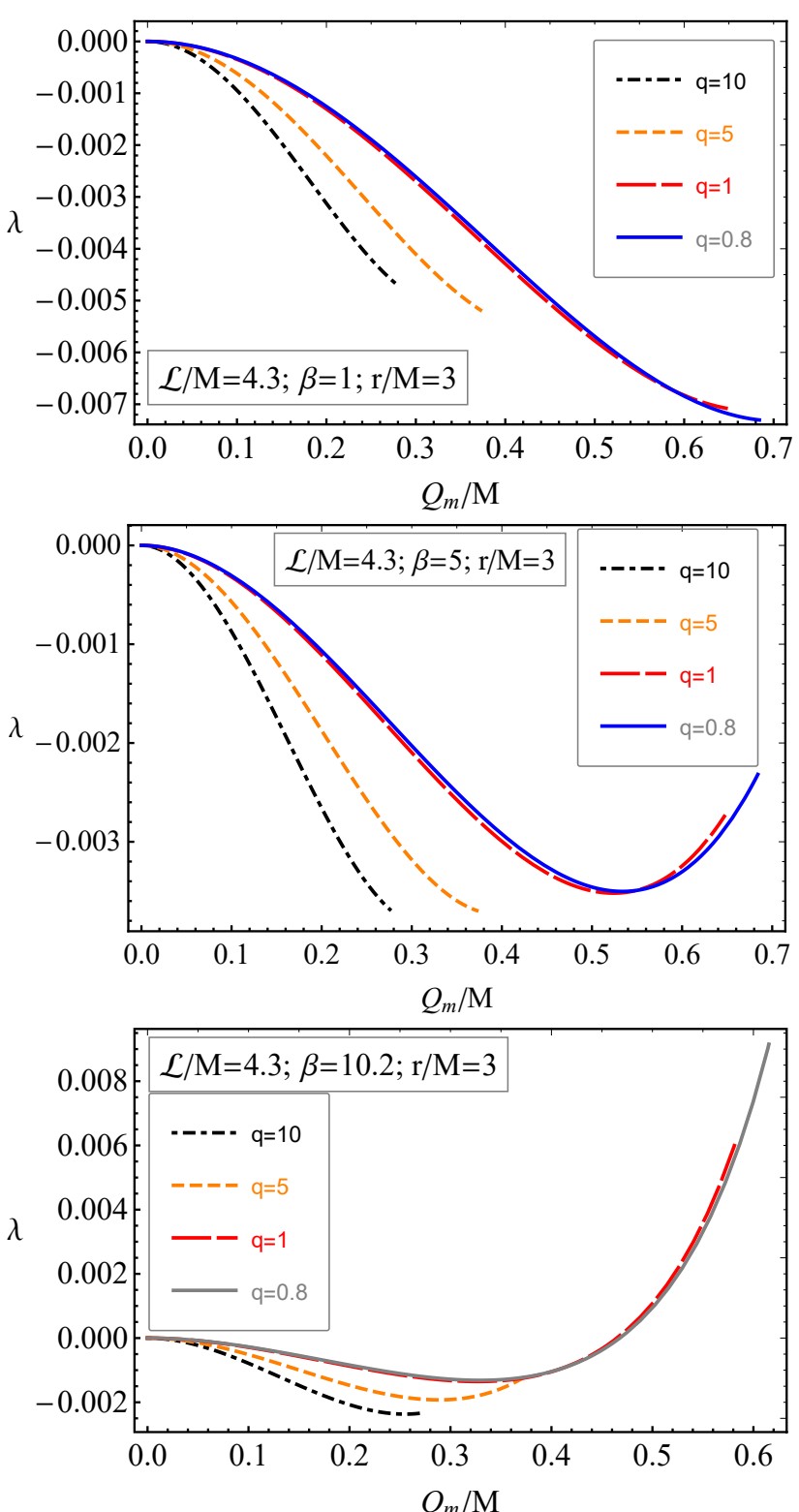

**Figure 10.** The dependence of Lyupanov exponent for magnetized particles from the magnetic charge of a regular nonminimal magnetic black hole for the different values of magnetic parameter $\beta$ at the distance $3M$ from the central BH. We use the value of the specific angular momentum of the particle, $\mathcal{L} = 4.3M$.

Plots in Figure 10 indicate that the Lyapunov exponent (LE) is negative for the magnetized particles with the smaller $\beta$ parameter, while for bigger values of the parameter $\beta$ and for the magnetic charge of the BH, the LE increases and takes positive values.

Now, we are interested in which values of $\beta$ parameter the LE vanishes. Here, we fix the magnetic charge as $Q = Q_{\text{extr}}$ and $\mathcal{L} = 4.3M$.

Now, we show in which values (the coupling parameter, the BH charge, and the parameter $\beta$) the LE equals zero, as shown in Figure 11. We find the coupling parameter of the Yang–Mills field, for which the LE becomes zero when the values of the BH charge are close to zero.

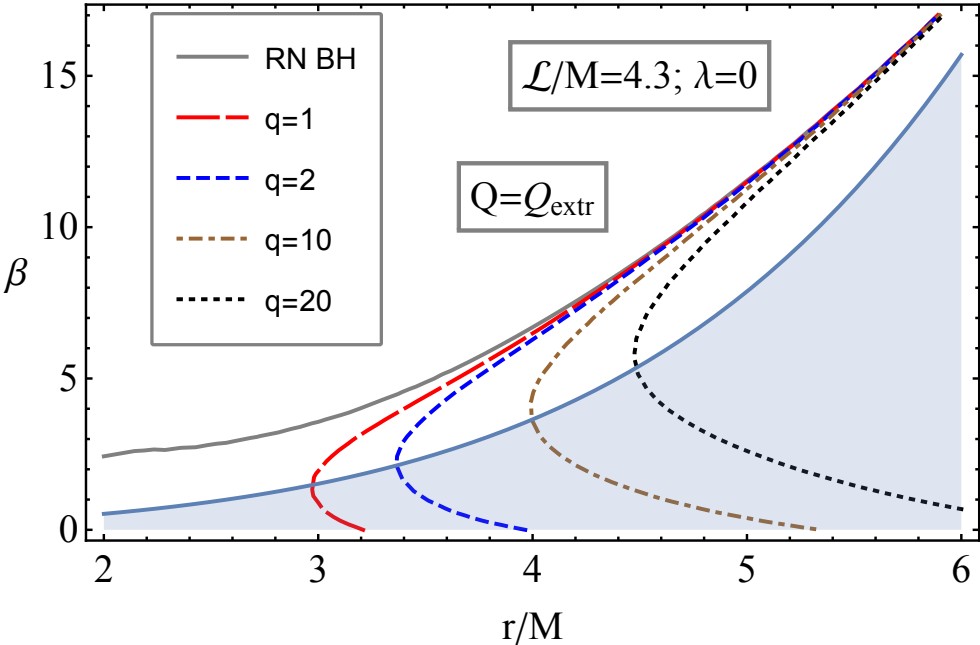

**Figure 11.** Relations between the parameter $\beta$ and distance, where the Lyapunov exponent for magnetized particles around extremely charged magnetic regular BHs is zero for the different values of the coupling parameter of the Yang–Mills field with the comparison to the extremely charged RN BH. Here, the coupling parameter, $q$ is given in the unit of $M^2$.

*3.5. Keplerian Frequency*

The angular velocity of particles measured by a distant observer or so-called Keplerian frequency is determined as [26,113,114]:

$$\Omega_K = \frac{d\phi}{dt} = \frac{\dot{\phi}}{\dot{t}} \, , \tag{31}$$

or explicitly,

$$
\begin{aligned}
\Omega_K = \frac{\left(2qQ_m^2 r + r^5\right)^{-1}}{\sqrt{\beta M Q_m + r^2}} &\left\{ r^8 \left( Mr - Q_m^2 \right) + 8\beta M q^2 Q_m^5 \right. \\
&+ \beta M Q_m r^6 \left[ r(2r - 5M) + 3Q_m^2 \right] + 2q Q_m^2 r^2 \\
&\left. \times \left[ r^2 \left( Q_m^2 - 3Mr \right) + \beta M Q_m \left\{ r(4r - M) + Q_m^2 \right\} \right] \right\}^{\frac{1}{2}} .
\end{aligned}
\tag{32}
$$

In further calculations, we convert the units of fundamental frequencies from geometrical (1/cm) to Hz (international unit systems, $s^{-1}$), which makes our analysis more understandable:

$$\nu = \frac{1}{2\pi}\frac{c^3}{GM}\Omega \, , \text{Hz} \, , \tag{33}$$

where $c = 3 \cdot 10^{10}$ cm/s and gravitational constant $G = 6.67 \cdot 10^{-8}$ $cm^3/(g \cdot s^2)$.

The radial dependence of the Keplerian frequency of test particles around a BH is shown in Figure 12. One can see from the bottom panel of the figure that the increase in the coupling parameter and the parameter $\beta$ causes the decrease in the Keplerian frequency.

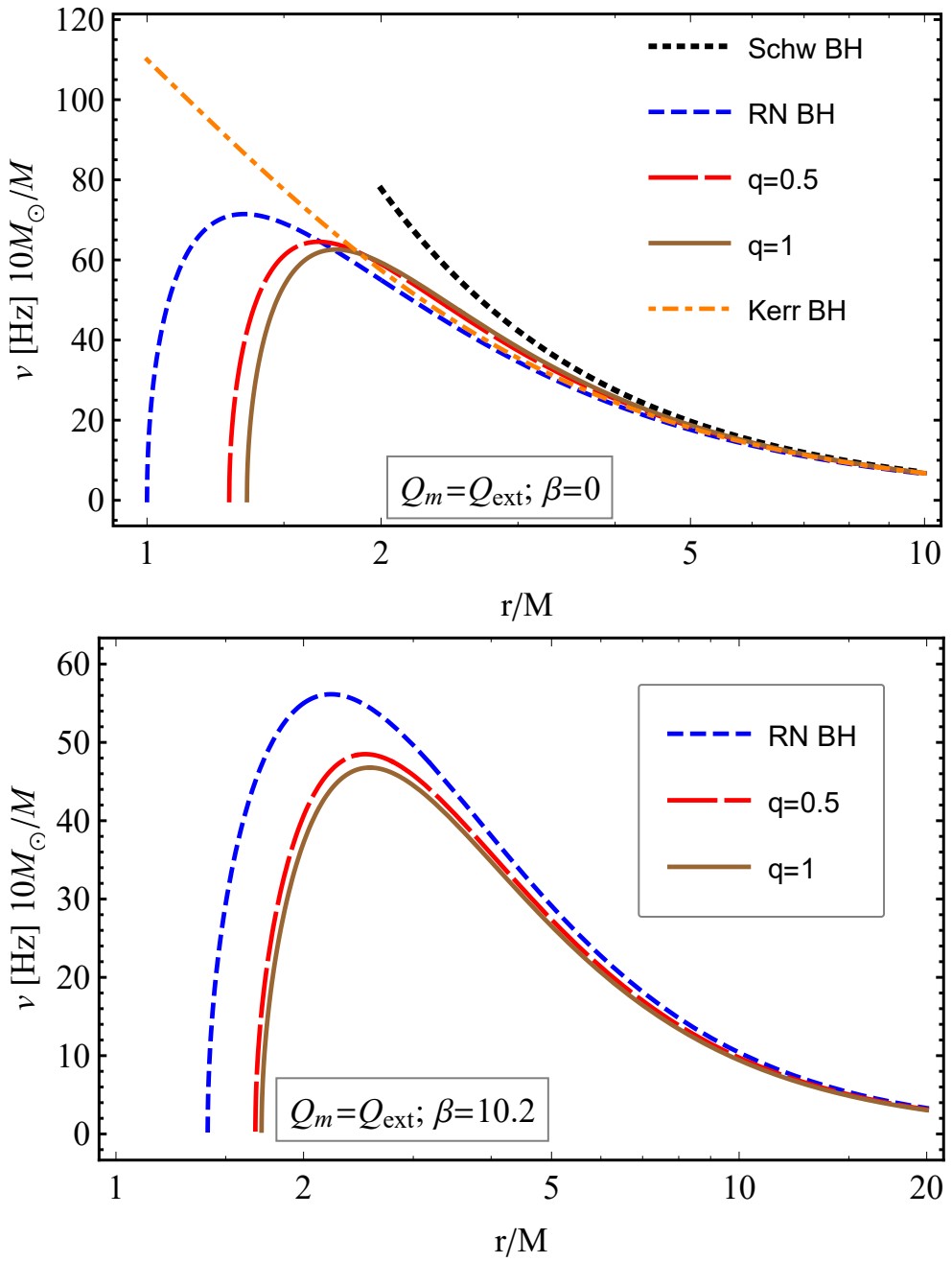

**Figure 12.** The radial dependence of the Keplerian frequencies of neutral and magnetized particles around extreme Kerr BH, extremely charged RN, and the nonminimal magnetic BHs for the different values of the nonminimal coupling parameter, *q*.

### 3.6. Epicyclic Motion of Test Magnetized Particles

In this subsection, we provide detailed analysis of the fundamental frequencies of the tested neutral particles moving around a BH. The effective potential can be expanded in terms of $r$ and $\theta$ in the form

$$
\begin{aligned}
V_{\text{eff}}(r,\theta) = {} & V_{\text{eff}}(r_0,\theta_0) + \partial_r V_{\text{eff}}(r,\theta)\Big|_{r_0,\theta_0} \delta r \\
& + \partial_\theta V_{\text{eff}}(r,\theta)\Big|_{r_0,\theta_0} \delta\theta + \partial_r \partial_\theta V_{\text{eff}}(r,\theta)\Big|_{r_0,\theta_0} \delta r\, \delta\theta \\
& + \frac{1}{2}\partial_r^2 V_{\text{eff}}(r,\theta)\Big|_{r_0,\theta_0} \delta r^2 + \frac{1}{2}\partial_\theta^2 V_{\text{eff}}(r,\theta)\Big|_{r_0,\theta_0} \delta\theta^2.
\end{aligned}
\tag{34}
$$

Here, we provide a careful analysis of this expansion, showing that the condition $\partial_r V_{\text{eff}} = 0$ makes the first term of Equation (34) zero, and the second term vanishes due to the stability conditions of the effective potential in the third and last terms of Equation (23). Thus, only two terms remain, which are proportional to the second-order derivatives of the effective potential with respect to the coordinates $r$ and $\theta$. The equation of motion in our calculations can be derived by replacing the derivation with respect to the affine parameter in Equation (23) into the time derivative, i.e., $dt/d\lambda = u^t$. This replacement allows one to obtain physical quantities measured by an observer at infinity. In order to obtain harmonic oscillator equations for displacements $\delta r$ and $\delta\theta$, we substitute Equation (34) into Equation (23) and follow all the above-mentioned facts [26,113]:

$$
\frac{d^2\delta r}{dt^2} + \Omega_r^2 \delta r = 0 , \qquad \frac{d^2\delta\theta}{dt^2} + \Omega_\theta^2 \delta\theta = 0 ,
\tag{35}
$$

where $\Omega_r$ and $\Omega_\theta$ are, respectively, the radial and vertical angular frequencies measured by a distant observer, defined as:

$$
\Omega_r^2 = -\frac{1}{2 g_{rr} \dot{t}^2} \partial_r^2 V_{\text{eff}}(r,\theta)\Big|_{\theta=\pi/2} ,
\tag{36}
$$

$$
\Omega_\theta^2 = -\frac{1}{2 g_{\theta\theta} \dot{t}^2} \partial_\theta^2 V_{\text{eff}}(r,\theta)\Big|_{\theta=\pi/2} .
\tag{37}
$$

Finally, expressions for the frequencies of the radial and vertical oscillations of magnetized particle take the following form:

$$
\begin{aligned}
\Omega_r = {} & \frac{(2qQ^2 + r^4)^{-4}}{r^2(r^4 - \beta^2 M^2 Q^2)} \Bigg\{ 2\Big(r^2\big(r(r-2M)+Q^2\big)+2qQ^2\Big)\Big\{ r^6\big[2qQ^2 r^3\big(MQ^2 + 18Mr^2 - 6r(M^2+2Q^2)\big) \\
& + 4q^2\big(4Q^6 - 15MQ^4 r\big) + r^6\big(9MQ^2 r + Mr^2(r-6M) - 4Q^4\big)\big\} + \beta^2 M^2 Q^2\big(2qQ^2 r^5\big[3MQ^2 - 42Mr^2 + 28Q^2 r \\
& + 12r^3 - 2M^2 r\big] + r^8\big[r^2\big(30M^2 - 21Mr + 4r^2\big) + Q^2 r(12r - 37M) + 12Q^4\big] + 4q^2 Q^4 r^3(11M + 12r) + 32q^3 Q^6\big)\big] \Bigg\} ,
\end{aligned}
\tag{38}
$$

$$
\Omega_\theta = \Omega_\phi = \Omega_K .
\tag{39}
$$

In fact, from Equation (38), it is difficult to see the effects of the magnetic coupling and magnetized parameters on the radial frequency. For this reason, we plot the equation by varying the parameters.

In Figure 13, we show the frequency of radial oscillations of the tested magnetized particles around the magnetic BH as a function of the radial coordinate. Here, we also show comparisons of the results with the case of an RN BH and the tested neutral particles. It is found that the frequency grows with the increase in the BH charge, nonminimal coupling, and magnetized parameters. It is seen that the effect of the magnetic parameter of a particle is stronger than the BH charge and nonminimal coupling parameters due to dominant effects of Lorentz forces.

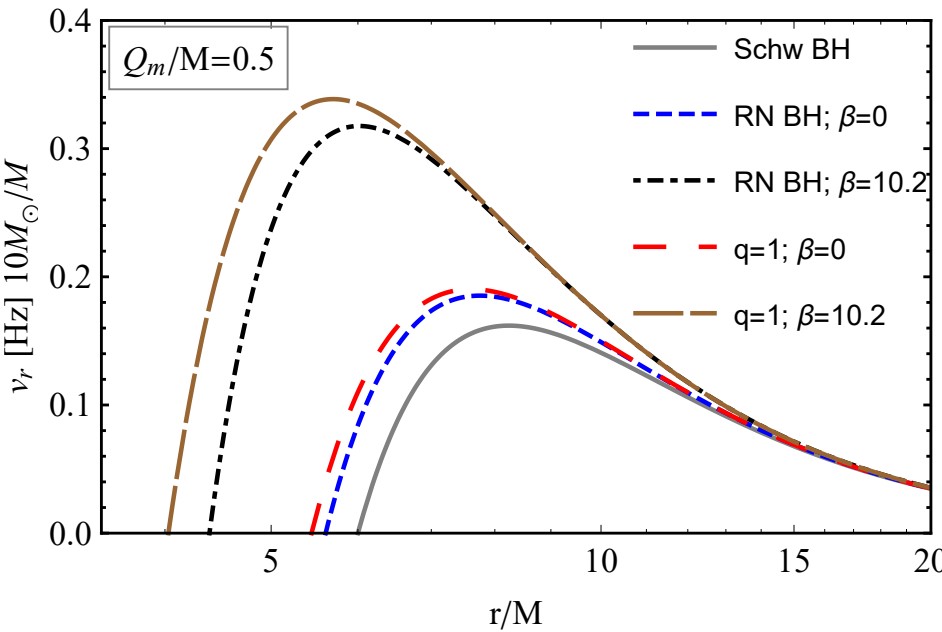

**Figure 13.** The radial dependence of frequencies of radial oscillations of test magnetized particles along stable orbits around a nonminimal magnetic BH for the different values of the nonminimal coupling parameter, $q$. The results are shown in comparison with the RN and Schwarzschild BH cases.

## 4. Magnetically Charged Particle Motion

This section is devoted to the magnetically charged particle motion around a regular nonminimal magnetic black hole. As the magnetically charged particle, we use a particle that has a non-zero magnetic monopole. The Hamilton–Jacobi equation of motion is one of the best tools to investigate the motion of magnetically charged particles in the axially symmetric spacetime of a regular nonminimal magnetic black hole. For the magnetically charged particle which is electrically neutral, the equation of motion reads as [87]:

$$g^{\alpha\beta}\left(\frac{\partial \mathcal{S}}{\partial x^\alpha} + ig A^\star_\alpha\right)\left(\frac{\partial \mathcal{S}}{\partial x^\beta} + ig A^\star_\beta\right) = -m^2, \tag{40}$$

where $g$ characterizes the magnetic charge of the test particle and $A^\star_\alpha$ is the dual vector potential with only one nonvanishing component, that has the form [87]:

$$A^\star_t = -\frac{iQ_m}{r}. \tag{41}$$

For the action (21), the explicit form of the equation of motion becomes

$$\frac{(2qQ_m^2 + r^4)\left(\mathcal{E} - \frac{gQ_m}{r}\right)^2}{r^2(r(r - 2M) + Q_m^2) + 2qQ_m^2} - \frac{\mathcal{L}^2\csc(\theta)}{r^2} \\ -\left[1 + \frac{r^2(Q_m^2 - 2Mr)}{2qQ_m^2 + r^4}\right]\left(\frac{\partial S_r}{dr}\right)^2 - \frac{1}{r^2}\left(\frac{\partial S_\theta}{d\theta}\right)^2 = 1, \tag{42}$$

with the specific magnetic charge of the test particle $g = q_m/m$.

The expression that comes from the equation of motion for the magnetically charged test particle moving at the equatorial plane ($\theta = \pi/2$) then takes the following form:

$$V_{\text{eff}} = \frac{gQ_m}{r} + \sqrt{\left(1 + \frac{\mathcal{L}^2}{r^2}\right)\left(1 + \frac{r^4\left(\frac{Q_m^2}{r^2} - \frac{2M}{r}\right)}{2qQ_m^2 + r^4}\right)}. \tag{43}$$

This effective potential has radial dependence, as shown in Figure 14. The top panel shows that the increase in the value of the magnetic charge of the test particle makes the effective potential stronger, which strengthens the attractive force between the black hole and test particle, which in turn makes the ISCO radius bigger (which is discussed later). The increase in the parameter $q$ in the middle plot shifts the maximum point to the left side that corresponds to the shift of minimum circular orbits for the given angular momentum and energy of the test particle. One can see in the last panel that the increase in these two parameters discussed makes the ISCO radius smaller, similar to the behaviour of the rotation parameter of the Kerr metric.

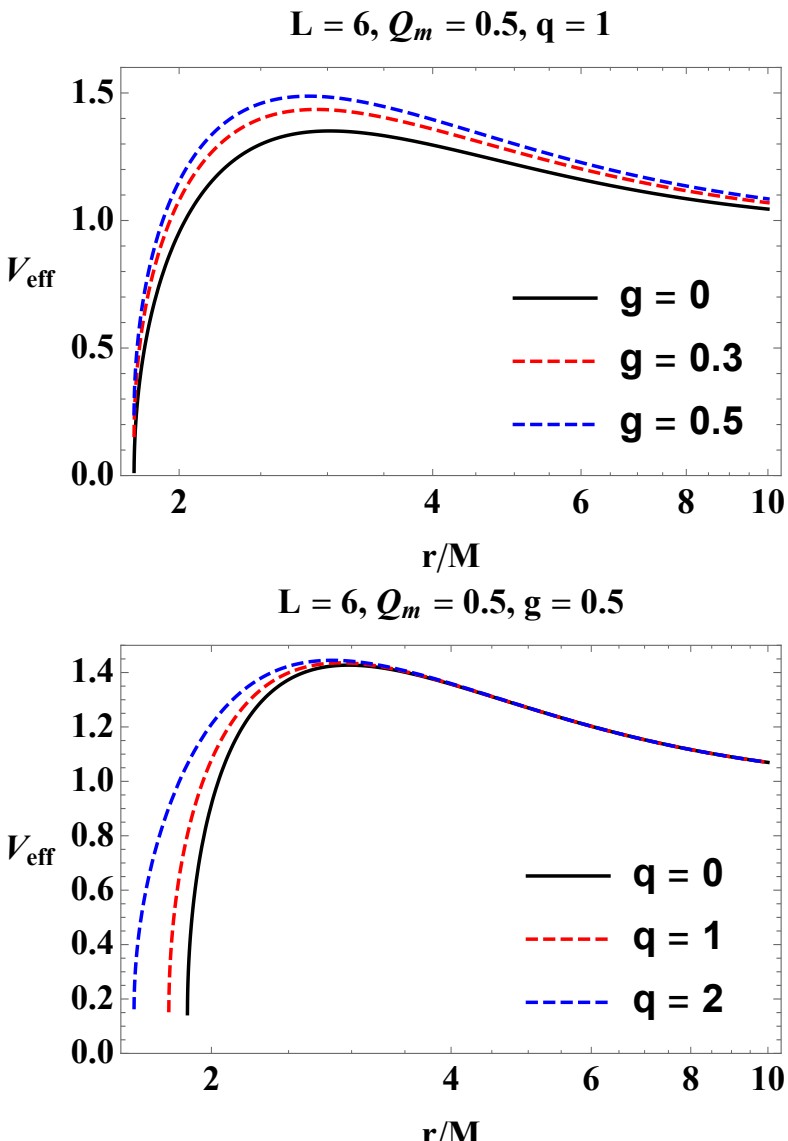

**Figure 14.** *Cont.*

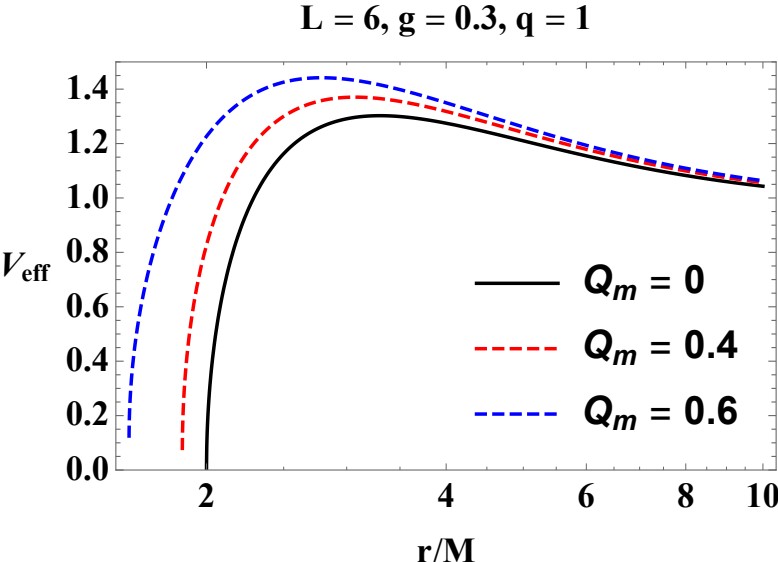

**Figure 14.** Effective potential of magnetically charged particle orbiting around a nonminimal magnetic regular black hole. Here, we provide units of the angular momentum and BH charge in $[L] = [Q_m] = M$, and the coupling parameters of Yang–Mills field in $[q] = M^2$.

From the well-known set of conditions on the effective potential given in Equation (26), one can find the dependence of the angular momentum of the particle from the circular orbit radius, as in Figure 15. These lines give us a more clear demonstration of the shift of the ISCO radius compared to the effective potential, since the minimum of these lines gives us the exact values of the ISCO radius for given values of black hole parameters and the magnetic charge of the test particle. From the first figure, it is apparent that the increase in the magnetic charge of the test particle shifts the ISCO towards bigger orbits. The second plot illustrates how the increase in the magnetic charge parameter of the nonminimal regular black hole makes the ISCO smaller. Furthermore, the last plot shows the shift of ISCO toward a small radius for bigger values of parameter $q$.

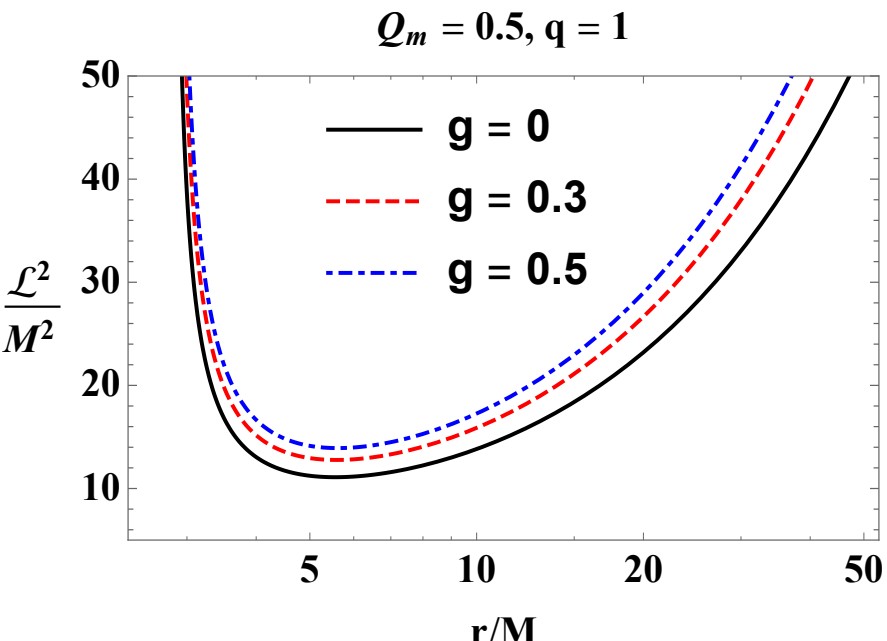

**Figure 15.** *Cont.*

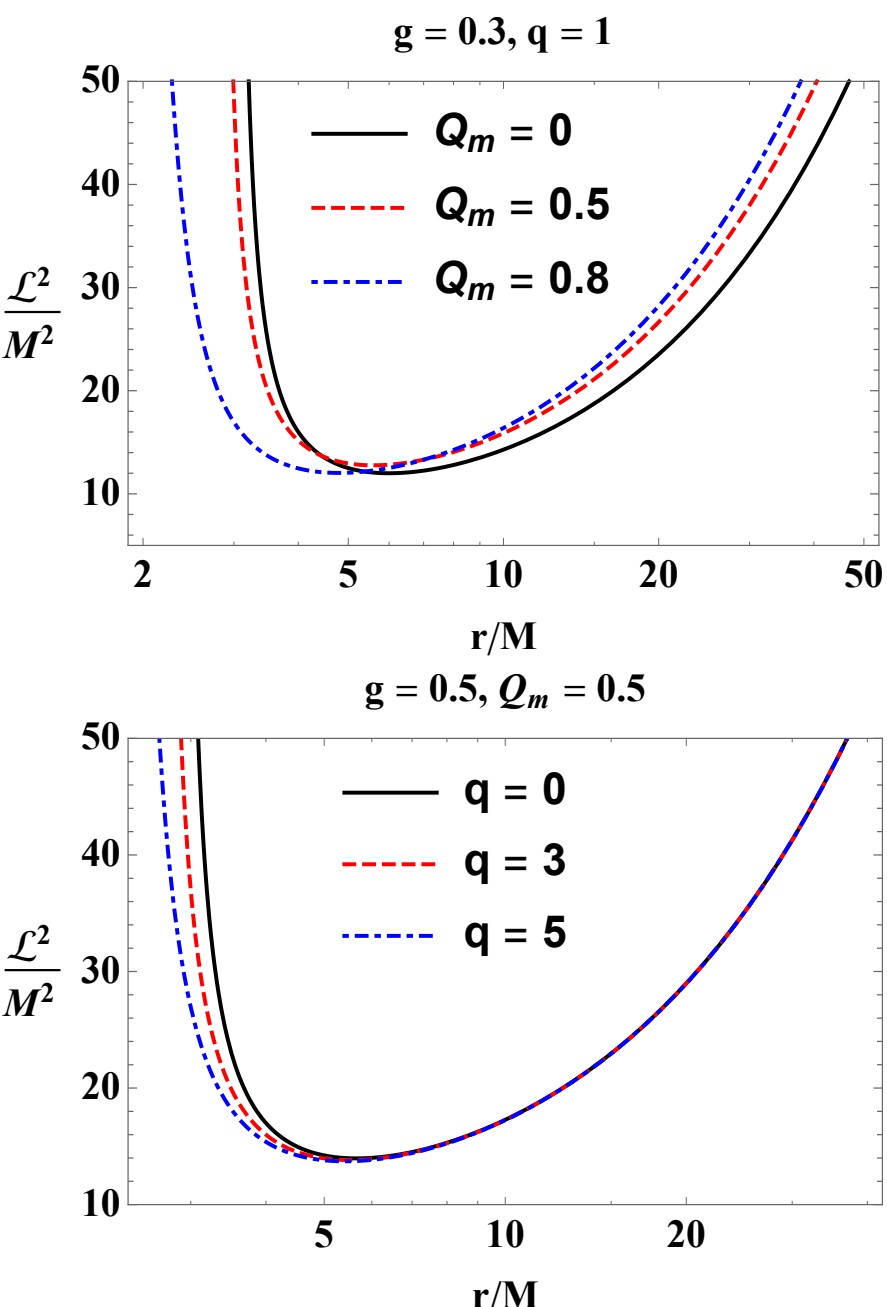

**Figure 15.** The radial dependence of angular momentum of the magnetically charged particle making circular revolutions on the equatorial plane of the nonimimal regular magnetic BH. Units of the angular momentum and BH charge and the coupling parameters of Yang–Mills field are taken as $[L] = [Q_m] = M$ and $[q] = M^2$, respectively.

Now, we turn to the investigation of ISCO for the motion of a magnetically charged test particle in the spacetime of a nonminimal regular black hole. From the additional condition on effective potential, $V_{\text{eff}}''(r) = 0$, or from the minimum condition for the angular momentum, one can easily find the relation between the ISCO radius and parameters of spacetime together with the magnetic charge of the test particle, which is shown in Figure 16. As was expected from the previous discussions, one can see how the ISCO radius behaves with the increase in the parameters mentioned. From the first figure, it is clearly seen that the increase in the magnetic charge of the test particle makes the ISCO radius bigger, which results in linear-like lines in the given range of the magnetic charge parameter of the test particle for chosen values of parameter $q$ and different values of

magnetic charge parameter $Q_m$ of the black hole. From the second and third plots, one can see how the ISCO radius decreases with the increase in the magnetic charge parameter of the black hole and the parameter $q$ for different values of the magnetic charge of the test particle.

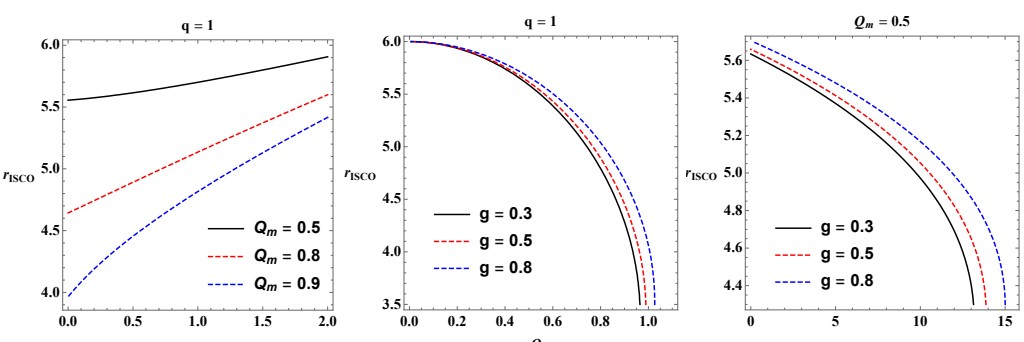

**Figure 16.** The dependence between ISCO radius of the magnetically charged test particle and parameters $Q_m$, $q$, and $g$.

Finally, we plan to see how well the parameters of spacetime, together with the magnetic charge of the test particle, can mimic the rotation parameter of the Kerr black hole. To do this, we follow the idea that if the black holes are believed to be nonminimal regular ones with the spacetime metric presented in this work but not the rotating Kerr one's, then the combination of these parameters should provide the same ISCO radius for the corresponding value of the rotation parameter. This idea leads to the investigation of the degeneracy between the rotation parameter of the Kerr metric and the parameters of interest for the matching values of the ISCO radius, which is illustrated in Figure 17. One can understand from the plots that the smaller the magnetic charge of the test particle we use, the better the mimicry that is obtained between the rotation parameter of the Kerr metric and magnetic charge of the nonminimal regular black hole and parameter $q$. From all three figures, it is easily seen that the middle one has better mimicry results compared to the other two, and it shows that for the given values of the magnetic charge of the test particle and parameter $q$, the magnetic charge of the nonminimal regular black hole can mimic the rotation parameter of the Kerr black hole up to the values of 0.8, which leads to the conclusion that black holes in the Universe that are characterized by the Kerr metric with the spin parameter up to this value can also be interpreted as effectively as the nonminimal regular BHs studied.

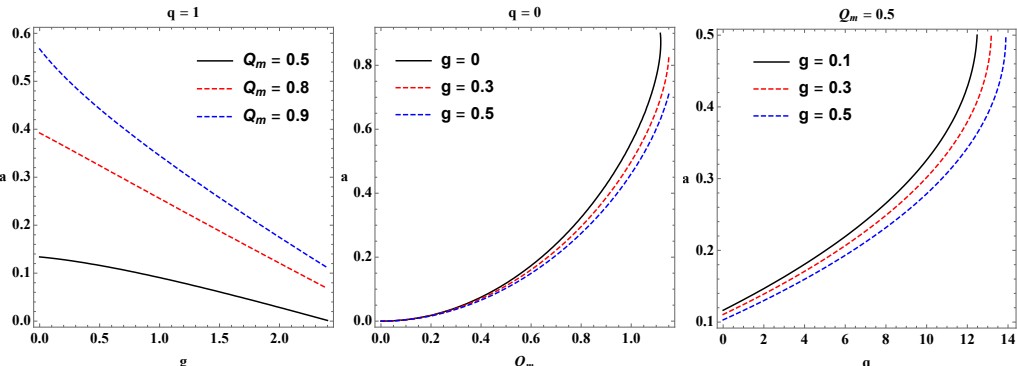

**Figure 17.** The degeneracy plot between rotation parameter $a$ and parameters $g$, $\Lambda$, and $q_m$, giving the same ISCO radius for the magnetically charged test particle orbiting around the regular black hole.

## 5. Conclusions

In the present work, we studied the effects of the Yang–Mills field on the event horizon of the spacetime around a regular nonminimal magnetic BH and the dynamics of

magnetized and magnetically charged particles. Our analysis shows that the minimal outer horizon which corresponds to the maximum value of the BH charge increases with the increase in the coupling parameter and reaches its maximum at $r_h = 1.5M$ when $q \to \infty$, while the maximum charge of the BH decreases and becomes zero.

We also explored the dynamics of magnetized particles around the regular magnetic BH. Note that we treated the magnetar SRG (PSR) J1745-2900 orbiting around the SMBH Sgr A* as a magnetized test particle and evaluated the magnetic parameter as $\beta = 10.2$. The performed analysis indicates that the presence of the BH charge and Yang–Mills field causes an increase in the specific angular momentum of the magnetized particles with the parameter $\beta = 10.2$ along circular orbits, while the energy decreases. We showed that the ISCO radii of the particles are very sensitive to the increase in the magnetic charge of the central BH. The ISCO radius decreases with the increase in the value of the BH charge. However, the increase in the Yang–Mills coupling parameter slightly decreases the ISCO radius with respect to the RN BH case. Here, we were interested in whether the BH charge could provide similar gravitational effects on the ISCO position as the spin of the Kerr BH. We showed that the RN BH can mimic the spin up to $a/M \simeq 0.7893$ and it is increased when the coupling parameter is $q = 1$ mimicking over $a/M = 0.82$ for the magnetized particles with the parameter $\beta = 10.2$. By studying the instability of orbits of the test magnetized particles with $\beta = 10.2$, we showed that higher values of the coupling parameter of Yang–Mills field make the orbits stable for the magnetic charge near its extreme value at $r = 3M$.

Moreover, we investigated magnetically charged particle motion in the spacetime of a static nonminimal regular black hole that has two extra parameters besides its mass. A study of the effective potential, angular momentum, and the ISCO showed that these parameters can mimic the role of the rotation parameter of the Kerr black hole. In turn, we studied how these parameters can mimic the latter one based on the idea that if black holes in the Universe can also be characterized by the alternate metric, then additional parameters included in this metric could provide the same ISCO radii as the spin parameter of the Kerr black hole. Our calculations showed that the given combinations of the parameters of the nonminimal regular black hole could mimic the spin parameter of the Kerr black hole up to around $a \approx 0.8$. This result led to the assumption that black holes with spin up to this value have a great interpretation with the spacetime metric of the nonminimal regular black hole for the motion of the magnetically charged test particle.

**Author Contributions:** Conceptualization, B.N. and J.R.; methodology, A.A.; software, B.N.; validation, B.N., A.A. and B.A.; formal analysis, B.N. and J.R.; investigation, B.N. and J.R.; resources, A.A.; data curation, B.A.; writing—original draft preparation, B.N. and J.R.; writing—review and editing, B.A. and A.A.; visualization, B.N. and J.R.; supervision, B.A. and A.A.; project administration, B.A. and A.A. All authors have read and agreed to the published version of the manuscript.

**Funding:** Ministry of Innovative Development of the Republic of Uzbekistan: MRB-AN-2019-29.

**Institutional Review Board Statement:** Not applicable.

**Informed Consent Statement:** Not applicable.

**Data Availability Statement:** Not applicable.

**Acknowledgments:** This work was supported by the National Natural Science Foundation of China (Grant No. U1531117). The research is supported in part by grants of the Uzbekistan Ministry for Innovative Development. A.A. is supported by the PIFI fund of the Chinese Academy of Sciences. J.R. acknowledges the ERASMUS+ project 608715-EPP-1-2019-1-UZ-EPPKA2-JP (SPACECOM).

**Conflicts of Interest:** The authors declare no conflict of interest.

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
