# Peer review of "Dynamics of Magnetized and Magnetically Charged Particles around Regular Nonminimal Magnetic Black Holes"

_galaxies, doi:10.3390/galaxies9040071_

Round 1

Reviewer 1 Report

The authors studied the dynamics of neutral and magnetically charged particles near a regular nonminimal magnetic black hole. They discussed how the coupling parameter $q$, magnetic charge $Q_m$, and parameter $\beta$ affect dynamics of particles, and then calculated the Lyapunov exponent to analyze the instability of circular motion. 

I think the following issues can be addressed:
1. The authors should first carefully polish the language and correct grammar mistakes. 

2. Eq.(22) and Eq.(26) are contradictory with each other. The authors should correct one of them.

3. The authors compared the role of the magnetic charge of the black hole with the spin of the Kerr black hole. But the details are omitted. Some details should be added.

4. Some citations should be added for Eqs. (30),(31),(35), etc.

Reviewer 2 Report

In this paper authors studied the motion of magnetically charged particles around a regular magnetic black holes. The spacetime investigated in this work is interesting and the analyses done by authors seems seems technically correct and interesting. However, I have few points related to this work:

  1. Authors refer to the solution given by Eqs. (4) and (5) as a regular magnetic black  hole. However, sometimes, in the literature the name 'monopole' is used. See example Ref. [25]. Although I do agree with the authors that this spacetime represents a black hole, authors should add a comment about this problem.
  2. Authors pointed out that this spacetime can mimic the spin parameter of the Kerr black hole, however this metric is most closely related to the RN black hole. Can we distinguish this spacetime from the RN spacetime with the present analyses? Authors should point out this problem and elaborate if needed.
  3.  In Eq. (14) authors consider the limit while taking the quantity 'q' to infinity. I suppose 'q' should be a small number, or at least, a finite quantity. Is such a limit physically acceptable? 
  4. Before Eq. (40) there should be a citation. 
  5. Page 9, subsection F, seem to me not completed. After Eq. (39) some discussion is needed. Note that authors here use the term 'harmonic oscillation', is this equivalent to the 'epicyclic motion' which is frequently used? 
  6. Some of the references need to be updated. For example, Ref [35] authors should add the complete reference: Phys. Rev. D 103, 024013 (2021).

Reviewer 3 Report

The Authors discuss the impact of magnetism on black holes with particular impact on the even horizon properties. This is timely due to the recent results of the EHT and the Authors contribution is to show that caution should be exerted in extracting informations and bounds on various models and variants of general relativity as similar effects can be mimicked by magnetic fields. The manuscript is nicely written but I kindly ask the Authors to check again the English, as there is potential for improvement. Among the points I spotted, here is a (partial) list: 'increases with the increasing the coupling parameter', 'supermassiv', 'in fact that', 'Riessner', 'allows to existence', 'of spin of cental black hole'.

Round 2

Reviewer 1 Report

The authors answered my questions and made proper corrections. Now I think this paper can be published in Galaxies.